# Sharing Reasoning Behind Individual Decisions to Invest in Joint Infrastructure

**Melle J. Nikkels** [1,2,3,*], **Joseph H. A. Guillaume** [4,5], **Peat Leith** [2] and **Petra J. G. J. Hellegers** [1]

1   Water Resources Management (WRM) Group, Wageningen University, P.O. Box 47, 6700 AA Wageningen, The Netherlands; petra.hellegers@wur.nl

2   Tasmanian Institute of Agriculture (TIA), University of Tasmania, Private Bag 98, Hobart TAS 7001, Australia; peat.leith@utas.edu.au

3   Aequator Groen & Ruimte, P.O. Box 1171, 3840 BD Harderwijk, The Netherlands

4   Water & Development Research Group, Aalto University, P.O. Box 15200, FIN-00076 Aalto, Finland; joseph.guillaume@aalto.fi

5   Fenner School of Environment and Society, Australian National University, Building 141, Linnaeus Way, ACT, Canberra 2601, Australia

*   Correspondence: melle.nikkels@wur.nl; Tel.: +31-620264171

**Abstract:** Development of joint irrigation infrastructure increasingly depends on investment decisions made by individual farmers. Farmers base their decisions to invest on their current knowledge and understanding. As irrigation infrastructure development is ultimately a group decision, it is beneficial if individuals have a common understanding of the various values at stake. Sharing the personal reasoning behind individual decisions is a promising approach to build such common understanding. This study demonstrates application of participatory crossover analysis at a workshop in Tasmania, Australia. The workshop gave farmers the opportunity to discuss their broader considerations in investment decisions, beyond just financial or monetary factors. It centered on the question, "In what conditions would you—the individual farmer—invest?" The participants' willingness to pay, in the form of crossover points, was presented as a set of scenarios to start an explorative discussion between irrigators and non-irrigators. Evaluation feedback indicates that the workshop enabled participants to share new information, improved understanding of differences between neighbors, and generated more respect for others and their decisions. As expected, reasoning went beyond economic concerns, and changed over time. Lifestyle choices, long-term intergenerational planning, perceived risks, and intrinsic motivations emerged as factors influencing water valuation. Simply having a facilitated discussion about the reasons underlying individuals' willingness to pay seems to be a useful tool for better informed decision-making about joint irrigation infrastructure, and is worth testing in further case studies.

**Keywords:** water valuation; participatory crossover analysis; irrigation; water resources management; willingness to pay (WTP)

## 1. Introduction

Building irrigation infrastructure is a long-term investment with the potential to transform a community. Decisions on the size and design of such infrastructure are critical and need to be made based on the best information possible—not just about water resources, but also regarding the values of the community. In the 20th century, decisions about irrigation infrastructure were dominated by cost-benefit analyses from the funders' perspective, with national governments and international donors particularly influential [1]. There is growing recognition that non-monetary aspects need greater consideration, and that stakeholder participation improves incorporation of these aspects [2–4].

In particular, we are seeing a trend whereby farmers co-invest with (local) governments to improve their water system [5–8]. Local stakeholders then need information to help them decide whether investment in an adaptive measure is "worth it". Relevant questions are, what information do they need and how do they get it?

In this paper, we first discuss existing techniques for valuation of water. We then briefly review the emerging concept of crossover point scenarios, and link it with water valuation. Crossover point scenarios depict conditions in which a decision would change (i.e., a crossover point occurs). Regarding irrigation infrastructure, such a crossover point scenario could be developed by asking individual farmers, "In what conditions would you invest?" Discussion of the answers to this question in a group setting could offer farmers the opportunity to share their broader set of personal considerations, beyond the financial or monetary.

We implemented the crossover point concept in a water valuation context by first eliciting individuals' willingness to pay (WTP), which then provided the starting point for a group discussion with peers. We made use of a case study area in Tasmania, Australia, where the recent design of an irrigation scheme was determined by individual farmers' decisions to invest in irrigation water during the construction phase. In a workshop setting, we used the value of water as a focal point for stimulating discussion of the reasons underlying individuals' WTP for irrigation water, and hence their decision to invest. Our ultimate aim was to enrich the participants' understanding of the value of water, so they could make better-informed investment decisions on irrigation infrastructure.

## 1.1. Valuation of Water

A farmer's decision to invest in joint irrigation infrastructure is based on their current knowledge and understanding of whether "benefits" will outweigh "costs". This can be either in the short term or in the long term, and based not only on monetary factors, but also, for example, on emotional and social factors [9]. Because benefits and costs include impacts on others and system feedbacks due to others' actions, the sharing of personal reasoning is expected to help farmers achieve a more comprehensive water valuation and build a common understanding of the factors that influence others' valuations of water [10]. Valuation is "the process of expressing a value of a particular object or action" [11,12]. Valuation happens both implicitly and explicitly and is at the core of investment decisions. Water valuation means "expressing the value of water-related goods and services in order to support their allocation and sharing" [13].

Valuation of water in agriculture is known to be problematic. Various authors have argued that such valuations may be biased or incomplete (e.g., [13–16]). This is because water is multifaceted, with economic, environmental, cultural, religious, and social dimensions [2,17]. In addition, values may change over time [18] and include both use values (e.g., irrigation) and non-use values (e.g., swimming and aesthetics) [14,19]. These multiple, personal and changing values of water are also discussed and reflected upon in the ecosystem services literature (see, e.g., [20–22]).

There is ample evidence [23,24] that investment decisions in water for irrigation deviate from those predicted by microeconomic models relying on a rational representative economic agent, maximizing its utility function under conditions of perfect information and in the absence of biases or unequal power relations [25]. Farmers have their own reasons, implicit and explicit, for making investment decisions [26]. The water valuation literature therefore suggests the need to empirically capture what stakeholders care about.

Indirect and direct inductive techniques are available to tackle this challenge and determine the value of water [14,27]. Indirect techniques deduce values based on observation, such as of market transactions, derived demand functions, and hedonic pricing. Indirect techniques can only provide estimated values and are considered suitable for valuing water resources that are marketed indirectly [15]. Direct valuation techniques, such as contingent valuation methods, elicit preferences directly by questioning individuals about their WTP for water [28]. Contingent valuation provides a means to gain insight into the personal values assigned to water in monetary terms [29]. It is not a

valuation of water itself (the so-called intrinsic value) [30]. Contingent valuation is widely applied to assess the monetary value of irrigation water (see [31–33]).

Hermans et al. [34] provides a broader view on value. These authors describe a method called "the mosaic of values", in which stakeholders and water professionals (in this case researchers) jointly assess the various values of water. They identify indicators for economic values, social values, and environmental values, and examine differences between farming systems. Water valuation with stakeholders thus becomes an avenue to share insights and incorporate the knowledge and expertise of participants. We agree with Hermans et al. [13] that, to support water resources management processes, existing valuation approaches need to be complemented with methods that bring stakeholders to center stage. Water valuation thus becomes part of a process in which stakeholders learn with and from each other. Placing stakeholders in such a central role responds to the call of the UN High Panel on Water [35] and Garrick et al. [2] for water valuation methods that incorporate the multiple and personal values of water.

*1.2. Crossover Point Scenarios and Discussion of WTP*

In parallel to the water valuation literature, there is a growing body of work on participatory methods to support water management processes [36,37]. For example, crossover point scenarios have been used to stimulate discussions in which participants learn from each other's perspectives on a particular problem. Crossover points are the conditions in which a decision would change. At that point, two alternatives would be equally preferable to an individual, for example, to buy or not to buy water; it is a point of indifference. Crossover points therefore offer natural scenarios for stimulating discussion about why a particular decision may be made [38]. Group discussions based on crossover point scenarios focus on two key questions: (i) when—in what conditions—does one alternative out-favor another; and (ii) what drives this (personal) preference [39]. Participants are encouraged to think beyond their day to day practice and ask themselves questions like the following: In what conditions would I change my opinion or preference, and no longer do what I am currently doing? Has that crossover point changed over time, and if so, why? What do I expect my future crossover point to be?

The sharing of personal perspectives in a group setting forms the core of social learning processes [40–42]. Recognizing this, the authors developed a workshop-based method called participatory crossover analysis (see [43]). The method engages participants in dialogue exploring the personal reasons underlying their preferences, to enable them to learn from and with each other.

With regard to valuation, WTP can be seen as a form of crossover point. A WTP scenario describes the financial or monetary conditions in which an individual's willingness to invest might change. Given the central role of valuations in investment decisions, the authors considered discussion of WTP scenarios in a workshop format to be a promising avenue to aid decisions on joint irrigation water infrastructure. In the discussion, participants could explore factors that might change their investment decision. Differences in crossover points within the group could provide the starting point for a facilitated dialogue about reasons and exploring where differences come from.

A focus on enabling dialogue could be a powerful tool for investigating the multiple and personal values of water, as called for by the UN High Panel on Water [35]. Such a dialogue could elicit non-monetary values, in addition to direct monetary costs and benefits, and then dig deeper to understand the assumptions and personal reasons underlying these values. By focusing on the processes that underlie different valuations, this approach builds on the contingent valuation literature, as recommended by Burgess et al. [44]. Rather than obtaining a single value estimate for an individual at a specific point in time, it promotes understanding of the factors leading to variation in values within a group and over time, anticipating non-stationary conditions over the service life of an irrigation scheme.

The dialogue facilitation approach is consistent with the grounded theory tradition [45], and avoids prematurely introducing a water valuation expert's preconceived notions. There is no predefined

model of value as there would be when starting from a cost-benefit analysis. There are not even predefined categories of costs or benefits, as there would be in a structured survey. Here it must be emphasized that the intention with this approach is not to find a "true" WTP or "right answer", as is often the aim of contingent valuation. Rather, a WTP is used purely as a starting point for discussion, providing participants an opportunity to explore their own reasoning and compare their ideas with those of others. The aim of the process is to give participants the opportunity to build confidence and capacity to make better-informed decisions.

This paper builds on previous work (in particular [38,39,43]) by explicitly implementing the participatory crossover analysis method, focused on WTP, in a water valuation setting. The case presented explored the potential for using a peer-to-peer learning workshop based on WTP scenarios as a means to elicit and share a rich set of considerations among a group of farmers (Dolinska and d'Aquino [46] further discuss the role of peer-to-peer learning workshops as spaces where norms are collectively constructed and new narratives can be produced) with the purpose of informing their decisions on investments in joint irrigation infrastructure.

## 2. Materials and Methods

### 2.1. Case Study Context

The Tasmanian government has set the ambitious goal of increasing the annual value of production in the agricultural sector from AU$1.8 billion in 2012 [47] to AU$10 billion by 2050 [48]. New irrigation schemes are to facilitate intensification and transformation of Tasmanian agriculture. Tradable water rights and user charges are being used as instruments to manage demand for irrigation water [49]. The irrigation schemes currently being developed and managed by Tasmanian Irrigation (TI) are being built to satisfy current demand in each region of Tasmania. Irrigation schemes are designed to deliver water at a reliability of at least 95% for a range of climatic scenarios [50] and to last 100 years. The approach to irrigation scheme design in Tasmania includes a pre-feasibility phase in which farmers must commit to buying water rights to cover at least 30% of the scheme's construction costs. The Australian Commonwealth and Tasmanian government cover the remaining 70%. The aggregated user commitments define the design of the scheme including the diameter of the irrigation pipes, and hence supply capacity [7].

The South-East Irrigation Scheme (SE3) provides the most expensive water of the state [51] and commenced operations in October 2015 [52]. SE3 serves agricultural enterprises around the townships of Tea Tree, Campania, Orielton, Pawleena, Penna, Sorell, and Forcett in south-eastern Tasmania (Figure 1). The yearly average rainfall here varies between 500 and 700 mm [53,54]. Current production is diverse, and includes annual crops (from barley to lettuces), perennial crops (from lucerne to cherries), and livestock (from wool to fattening organic cattle). The nearby Hobart airport caters for high-value export crops.

SE3 water is sourced from the Derwent River, approximately 30 km from the first outlet. A pipeline distribution network delivers pressurized, almost drinking quality water throughout the SE3 district. According to a TI irrigation scheme manager, almost 400 landowners were contacted during the SE3 2012 water sales period. By January 2018, 62 water entitlements had been sold. Based on the aggregated commitments in the pre-feasibility stage, the SE3 scheme now has the capacity to supply 3000 ML ($3 \times 10^9$ L) of water during a 180-day summer delivery period (October–March). In addition, 3000 ML can be supplied during a 180-day winter delivery period (April–September). At the time of this writing, a limited number of summer water rights was still available in parts of the scheme (dependent on location and pipe diameter), but no winter rights were being offered for sale as yet. The cost of a water right has a one-off component, currently set at AU$2700/ML; a yearly fixed component of AU$140/ML; and a variable delivery component of AU$178–AU$220/ML [51].

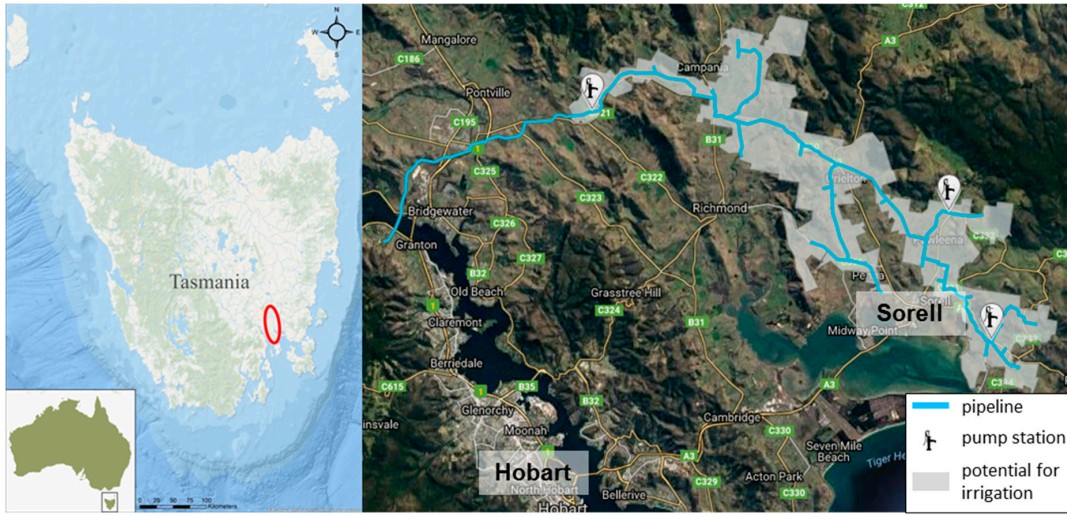

**Figure 1.** Case study area of the South-East Irrigation Scheme Phase 3 (SE3) district in south-east Tasmania, Australia. The SE3 image was adapted from [52].

In the adjacent Coal River Valley, farmers from different backgrounds are united in a community of practice which gathers regularly (www.coalriverproducts.com.au). Connecting with peers is widely acknowledged as having a positive influence on learning [55,56]. In the SE3 district, there is no such formal platform or association in which irrigators and non-irrigators exchange ideas and learn from each other. There are various industry associations (e.g., Wine Tasmania, Fruit Growers Tasmania), but these bring together farmers within the same business and lack a district focus. There is, however, an irrigation committee, which represents irrigators in meetings with TI. After a workshop organized in February 2017 with farmers in the Coal River Valley, participants expressed curiosity about the reasoning underlying their neighbors' investment decisions, as all agreed it would be impossible to make a profit from either livestock or traditional annual crops (e.g., cereals) using SE3 water. Farmers would have to switch to high-value cropping enterprises [39]. This was backed by a cost-benefit analysis provided by TI to potential buyers. It indicated that the yearly cost of SE3 water could not be recouped if used to irrigate pasture to finish store lambs and cattle or to grow cereals under irrigation [57]. However, when cycling or driving through the area one observes both cereals and pasture (for grazing) under irrigation. This suggests the hypothesis that farmers' personal reasons for investing in the irrigation infrastructure go beyond short-term economic gains, as suggested by Raworth [25]. This hypothesis has implications for the information provided to farmers to assist them in their investment decisions on joint irrigation infrastructure. In the water-related outcomes of the workshop, we therefore look closer at the personal reasons individual farmers gave for deviating (or not) from the recommendations of the cost-benefit analysis in their decision to invest in the joint irrigation infrastructure and their use of SE3 water.

*2.2. The Workshop*

The workshop was structured according to the five-step participatory crossover analysis framework proposed in [39]. The first step was verification that the workshop's aims were consistent with the aims of participatory crossover analysis, namely, to elicit personal reasoning, to improve understanding of where differences in preferences come from, and to gain insights for regional planning.

The second step was to examine the decision-making context and determine whether suitable conditions could be provided for a participatory crossover analysis workshop. Those conditions are four: (i) preferences must be subjective, meaning there is no objective optimum; (ii) there must be at least two discrete alternatives to compare; (iii) there must be a dialogue situation; and (iv) a capable facilitator must be on hand. In the SE3 scheme, the first two conditions were taken to be met,

as investment decisions were non-uniform at the SE3 pre-feasibility stage. Moreover, with the sale of summer rights under way and that of winter rights expected to begin within two years, there were still relevant alternatives to discuss. Regarding (iii), we assessed the dialogue situation during on-farm interviews (described in Section 2.2.2). For a constructive dialogue, participants must be willing to listen, have the ability to provide or explore an explanation for their position, and must be open to reflection [58,59]. With respect to (iv), the first author had previously facilitated a crossover workshop in the adjacent valley and was familiar with the farming context.

Steps 3 and 4 of participatory crossover analysis, respectively, the process of preparing for and organizing the workshop, are presented along with the on-farm interviews in Sections 2.2.1–2.2.3. The evaluation (step 5) is presented in Section 2.2.4.

### 2.2.1. Selection of Participants

As there was no existing community of practice, we contacted the chairperson of the irrigation committee for contact details of both irrigators and non-irrigators covering as diverse a set of enterprises as possible. This selection procedure resulted in a list of thirteen possible attendees for the workshop: two perennial fruit growers, one lettuce grower, five irrigators who at the time used irrigation waters for grains and pasture, four non-irrigators, and one investor. There was a wide range in age and farm size, but just one female farmer. All prospective attendees had long-term farming experience. The chairperson contacted them to secure their permission, after which we started the initial phone enquiry to explain the process. The process entailed a farm interview and a workshop, with participants allowed to withdraw at any stage. Twelve of the thirteen possible participants agreed to be interviewed; only the investor was not interested. A Dutch researcher with an agricultural background (first author) conducted the farm interviews in November and December 2017. This was the same person who facilitated the workshop.

### 2.2.2. Interviews

The 1–2-h interviews had a structured part and a semi-structured part. The interviewer began with the structured part, asking descriptive questions about farm and water use characteristics to explore the differences between enterprises (Table 1). The semi-structured part then focused on the personal reasoning underlying investments in irrigation and views on future valley development (see Appendix A for an interview script). Afterwards, the interviewer asked clarification and follow-up questions and used summarizing to dig deeper into interviewees' reasons, as recommended by Dunn [60]. The personal views and insights thus obtained were used to develop the relevant workshop questions and follow-ups. The interviews were also used to introduce the idea of crossover points as WTP. The interview process, moreover, allowed participants to get to know the facilitator, build a relationship, and make an informed judgement about whether the workshop would be worth their time. To avoid interrupting the "flow" of the conversation by note-taking, the interviews were recorded.

**Table 1.** Interview findings on farm characteristics, including water use and additional costs for irrigation.

|  | Min. | Average | Max. |
|---|---|---|---|
| **Total hectares** | 30 | 243 | 600 |
| **Hectares under irrigation** | 0 | 23 | 70 |
| **Hectares under irrigation in the future** | 0 | 70 | 200 |
| **Years of experience with irrigation** | 0 | 15 | 40 |
| **SE3 water allocation (ML)** | 0 | 65 | 300 |
| **Current water use (ML)** | 0 | 47 | 200 |
| **Water use per ha (ML/ha)** | 0.6 | 2.1 | 5 |
| **Current value generation ($/ML)** | 0 | 3277 | 20,000 |
| **Pumping costs ($/ML)** | 0 | 89 | 120 |
| **Other capital costs to start irrigating ($/farm)** | 0 | 348,750 | 2,000,000 |

During the interviews, both irrigators and non-irrigators showed signs of not really being open to input from others. Illustrative here is the comment, *"The others just don't have a clue"*. Some said the scheme had divided farmers into an irrigator group and a non-irrigator group, leading to a polarized situation. Thus, the conditions for a constructive dialogue seemed to be lacking. Instead of not proceeding, we decided to try to create conditions conducive to a dialogue during the workshop.

### 2.2.3. Workshop Design

For both the process and content, workshop design is of major importance [36,61]. Location, duration, and number of participants are all design decisions that influence the discussion process [62,63]. Our three-hour workshop was held at the Sorrell Community Center in December 2017 (summer). The fact that the situation was initially polarized was underpinned by overheard comments such as, *"What is he doing here? He knows nothing about farming"* and *"Luckily he was not invited, he would not have added to the discussion"*. Four irrigators and three non-irrigators attended. With the permission of the participants, the workshop was recorded and transcribed verbatim. In addition, a note-taker assessed participants' reactions during the workshop process and assisted the facilitator by indicating when clarification was needed.

Workshop facilitation, tools, and establishment of ground rules are known to influence the discussion process [36,62]. When opening the workshop, the facilitator thus spent time outlining the conditions required for a constructive dialogue. To avoid being demanding or directive, the facilitator focused on the opportunity the workshop provided for participants to learn about the personal reasons of others through listening, sharing, and being open to differences. The facilitator explained they would discuss hypothetical scenarios, in which each participant had their current knowledge and experience with irrigation, but assuming no irrigation infrastructure was yet in place and a "once in a lifetime opportunity" to buy water was on offer, as it had been during the pre-feasibility stage of SE3.

To gauge and display the WTP, we used a PowerPoint add-in for polling called TurningPoint [64]. This includes personal clicker devices that allow participants to provide anonymous answers. Four questions—checked for relevance by a non-attending member of the Irrigation Committee and a TI scheme operator—guided the workshop:

1. What is the maximum price per ML that you would be willing to pay for a water right? Why?
2. What is the maximum price you would have been willing to pay for a water right if reliability had been 80% (instead of 95%)? Why?
3. What is the maximum price that you are willing to pay for winter water rights ($/ML)? Why?
4. What is the minimum value/ML you need to generate to make SE3 water worthwhile? Why?

Participants were asked at what specific price their preference would change and to compare alternatives. The wording of these questions was thus consistent with standard practice in WTP analyses (see, e.g., [65]). Participants were asked to fill in their initial WTP by clicking their personal devices. Additionally, they were asked to indicate how confident they were about their WTP on a paper worksheet. The facilitator displayed the anonymous answers. To start the discussion, the facilitator asked for a volunteer to explain their reasoning, without necessarily identifying which WTP response was theirs. From this initial personal account, other participants added to the discussion, saying why their WTP differed (or not). The facilitator closed each topic by asking the participants to again fill in their WTP and indicate their confidence level on the worksheet. In addition, participants were asked to record on the worksheet whether their answer had changed and if so, why. This process was repeated for each question. For the workshop script, see Appendix B.

### 2.2.4. Evaluation

Directly after the discussions, the note-taker facilitated a thirty-minute evaluation to gather preliminary feedback on the workshop. Topics addressed were whether participants had been

comfortable sharing their reasoning, whether they believed others in the group were being honest during the workshop, and whether they would recommend the workshop to others.

A more detailed evaluation was conducted in the weeks following the workshop, focused on learning-related outcomes. By phone, each participant was asked open questions including what they remembered as particularly useful or interesting, if and how the workshop had changed their thinking and decision-making, and the perceived usefulness and value of the discussions to themselves and to others (Appendix C). A previous study [39] found that participants may have difficulty making explicit "what" they learned and how it might influence future investment decisions. We therefore also sought to capture the perceived usefulness of the event by asking about participants' willingness to recommend the workshop to others and to participate in a future workshop [62].

## 3. Results

### 3.1. Workshop Discussions

Before starting the discussions, the facilitator displayed the farm characteristics findings from the interviews (Table 1). In addition to the cost of the water rights, farmers opting to start irrigation faced other costs, such as for pumps, pipes, irrigators, farm dams, planting trees, and landscaping. These additional costs ranged from zero to two million Australian dollars per farm. Workshop participants briefly discussed these differences. Table 1 shows that the valley is still in transition as, currently, not all water is being used. Most of the irrigators interviewed said that they were still in the process of deciding what they wanted to use the irrigation water for, and they expected to increase both their area under irrigation and water use in the future.

### 3.1.1. WTP for SE3 Water

The first workshop question elicited the WTP for a right to one ML of water. The participant starting the discussion on the question of "why?" said that their WTP hinged on more than just economic reasoning, and qualitative, personal factors were actually central (see quote 1 in Box 1). The cost-benefit analysis provided by TI [57] did guide the decision rule of this participant. Non-monetary factors had more "weight". Other participants agreed with that statement, especially expressing a preference for buying land where it rains instead of buying water.

Then one of the irrigators explained that he had invested due to the climate and location (quote 2, Box 1), but said that he could see why other participants had made a different choice. From here on—relatively soon after posing the first discussion question—participants acknowledged differences among themselves. They asked each other questions for clarification, patiently explained, and treated each other with respect—while engaging in a lively dialogue.

The conversation then touched on the difference between actual uptake and what they would have liked to buy, as some participants had temporary budgetary constraints. Others wished they had used a different bank, as they could have borrowed more. None of the irrigators indicated they had too much water. Indeed, two had bought more water on the water market since they started irrigating, and others were considering increasing their summer allocation.

**Box 1.** Quotes regarding WTP for water.

| | |
|---|---|
| 1. | *"You have to look at the infrastructure cost, you have got to look at your age, and you have to look at the quality of life. I went to a lot of meetings and people were not there because they had to move irrigators and were up in the middle of the night. I am more comfortable and have less stress while doing what I am doing."* |
| 2. | *"In my case, this is the ideal place. It has the climate to do what I want to do so I am prepared to pay for it. I have a business that needs the water. But I can appreciate their view."* |

3.1.2. Influence of Reliability and Supply Regime (Winter/Summer) on WTP

When discussing the influences of reliability on their WTP for water, it became clear that not all benefits were tangible and that they may vary over time. One participant said they used water to "drought proof" the core of their business, reducing their own stress level (quote 3, Box 2). An irrigated part of the business can feed into other parts of the business, this farmer explained, reducing weather-related risks and allowing for robustness and flexibility to wait for better market prices. A lower reliability would influence this drought-proofing function and therefore the stress-reduction impact of owning irrigation water rights.

What participants perceived as a risk turned out to be personal, as some said they saw investing in water as a capital risk—due to the need for mortgages and employment contracts. This perception of risk, which seemed partly related to reliability, influenced their WTP, as displayed in Figure 2. Two participants indicated that they would not pay for water with a reliability of 80%.

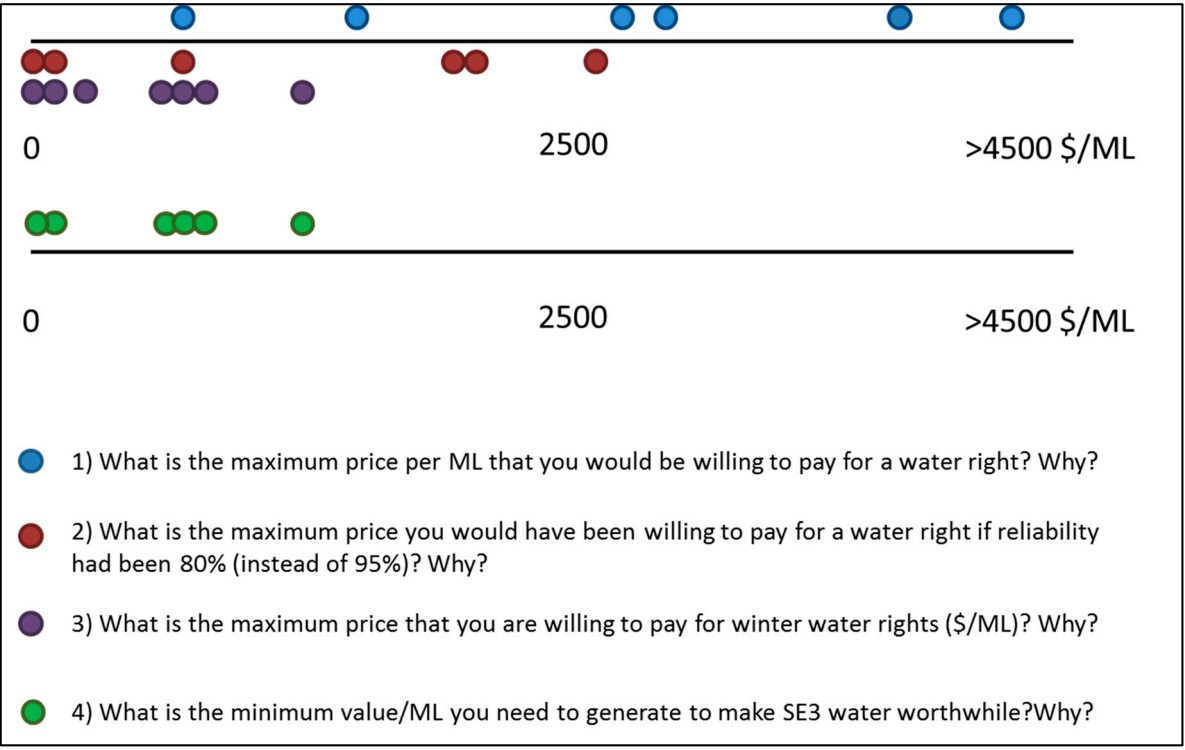

**Figure 2.** Responses to each question after discussion: Willingness to pay and minimum value generation.

When discussing winter water, participants noted the extra costs for water storage in the winter and evaporation losses if water is only needed in summer. They therefore agreed that winter water should be cheaper than summer water, which was reflected in their WTP. For all this was less than AU\$2000/ML. The irrigators also argued that they should be given the opportunity to buy winter water rights cheaper than "outsiders", as without their investment in summer water, the scheme would not have been built. They heard that TI was considering the same price for winter water as for summer water (AU\$2700/ML), which they perceived as unfair. Both irrigators and (current) non-irrigators seemed to agree that demand for water was growing, but that it would take a long time to sell winter water at \$2700/ML. One participant indicated it would take at least 20 years, with properties being sold to new owners, before all winter water could be allocated at that price.

3.1.3. When is Water Worth it?

In the discussion about what minimum value/ML a farmer needed to generate to make SE3 water "worth it", the first discussant suggested a 10% rule of thumb. This participant saw investment in

water as an input that needed to lead directly to an increased return, in terms of profit, instead of focusing on value added to the farm enterprise overall (quote 4, Box 2).

In the ensuing discussion, others noted that their land purchases did not achieve a 10% return. They saw an investment in water rights as similar to an investment in land: slowly increasing in value, and therefore not a cost, but a good long-term capital investment (quote 5, Box 2). As a long-term capital investment, participation in the water scheme was also said to be a way to prepare their enterprise for generations to come.

For some participants, irrigation seemed to be part of their identity; a lifestyle. Among this group, the decision to invest in irrigation was not solely based on making profits, but involved personal intrinsic motivations (quote 6, Box 2).

**Box 2.** Quotes illustrating non-monetary factors influencing WTP.

| | |
|---|---|
| 3. | *"Not F5-ing (refreshing) the weather page anymore."* |
| 4. | *"I look at it in a different way. I just look at the profit and I think you have to generate a 10% return on that extra capital. I am not sure how much extra value that generates, but profit is what drives me. That figure is around the 10% mark. You have to make at least $300/ML extra profit, every year, to make it worthwhile."* |
| 5. | *"If we did not buy it, the scheme would not have been in. My children might want to go on and grow cherries or other horticulture."* |
| 6. | *"I spend a lot on irrigation. Probably don't make a lot of money, but I love doing it. I love seeing the crops. Getting them up and growing. You know, some people might think that I am crazy but it is my thing. I don't go out in a boat or anything else. This is my life."* |

### 3.2. Evaluation Results

The workshop was clearly successful in stimulating farmers to share their personal reasoning. As expected, the discussions went beyond financial and monetary factors. Survey results (Table 2) indicated that all participants felt comfortable participating and were satisfied with the facilitation.

In their immediate evaluation response, the majority of participants was also positive about the workshop as a whole, but with some hesitation about whether they would recommend it to others. However, in the follow-up phone interviews, all said they would recommend the workshop to other irrigator groups, especially during the pre-feasibility stage of building a new irrigation scheme. At the end of the workshop, participants suggested organizing another workshop to share their understanding of the value of water for irrigation with farmers from the Coal River Valley, water managers, and policymakers. That workshop took place in May 2018; results are presented in forthcoming work.

Interestingly, the value of the workshop emerged on reflection rather than immediately. Comments immediately after the workshop and in the follow-up phone interviews indicated that some had been disappointed that there were just four questions, and felt they could have covered more questions. Others pointed out that these four questions had kept them entertained for more than two hours. One participant raised the question, "*What did we learn tonight?*" After the facilitator recapped the workshop's aim as being to enable the participants to get a better understanding of the differences in farmers' valuations of water by focusing on personal reasoning, they seemed more satisfied with the outcome.

While, for the most part, the participants could not directly link what they learned to specific points in the workshop, they acknowledged the value of the discussions. Participants also acknowledged differences within the group as an asset that could benefit future dialogues, learning, and decision-making. In addition, participants' attitudes seemed to have changed. They acknowledged the perspectives of others, which is a strong indicator of learning from and with peers [62]. Participants said they might still disagree, but they respected each other more (quotes 7–9, Box 3).

While positive about the workshop, participants had a mixed view of the added value of the WTP questions, as is to be expected at such a proof-of-concept stage. Most participants were not surprised by the preferences expressed by others—though these were still a source of learning for

some. This was anticipated, given that WTP was used as a "straw man". The workshop's intent was that the discussion following the first "guess" would be perceived as more insightful than the WTP itself. Some did find the framing—using WTP—to be useful, suggesting it would be worthwhile for future work to investigate factors affecting perceptions of utility. In this regard, evaluation interviews provide evidence for two key hypotheses.

First, the perceived utility of the method was influenced by its execution. The task of expressing confidence levels in particular was said to be "*a bit confusing*", "*a bit tedious*", and "*could have been more straightforward*" (quotes 10–12, Box 3). At the same time, there was recognition that this perception might have been different for others (quote 13).

A second hypothesis is that it was difficult for participants to evaluate the value of the method at the time of the survey, especially in terms of its impact on group understanding. In the follow-up interviews, participants recognized the need to cater for other participants' needs (quote 14). The fact that the role of the WTP was deliberately transient made it difficult to reflect on its impact—though participants did acknowledge that it successfully stimulated discussion (quote 12, 15). These issues could be addressed in a more rigorous evaluation of the impact of using the WTP to start the discussions.

**Box 3.** Quotes reflecting on the value of the workshop and the focus on reasons underlying WTP.

| | |
|---|---|
| 7. | "*I have been psychoanalyzing the district for years. I know everyone's background. I studied it and I was not surprised about others' preferences. It is really about respecting each other's decision. Is that not funny? I have never confronted others and asked them why they did or did not buy water. But for the first time today, with everyone explaining their personal reasons it makes a lot of sense. It is just the reason "why" that makes me accept everybody's preferences. Before I thought they were so silly, but now it does not seem silly: They have realistic reasons and I respect that now.*" |
| 8. | "*Everyone has their reasons for what they want or don't want to do and they are welcome for that.*" |
| 9. | "*Everyone here is so different and you can't go out and judge everyone.*" |
| 10. | "*The piece of paper with confidence levels was a bit confusing for everyone I think.*" |
| 11. | "*It could have been more straightforward. It could have been a bit more simplified.*" |
| 12. | "*I found it a bit tedious writing those things down, whatever we had to write down, the confidence. I found that pretty annoying. But all the verbal stuff was good. I did not mind pressing the button on the screen. I thought that was quite clever. It brought everyone's thoughts in but it did not name any names. It was just a trend of what everyone was thinking.*" |
| 13. | "*It was easy for some but other people still need to understand it too.*" |
| 14. | "*It took a long time. I already thought about it myself and had an understanding of the other perspectives and so I personally did not need three hours to discuss it, but others did seem to need that time.*" |
| 15. | "*I am a little bit skeptical about just taking an arbitrary value and selecting that for the questions we have been through. It certainly has prompted discussion and thinking about the subject. So it has added something.*" |

**Table 2.** Results from the evaluation immediately after the workshop.

| Environment | | | | | |
|---|---|---|---|---|---|
| | Strongly disagree | Disagree | Neither agree nor disagree | Agree | Strongly agree |
| I believe others in the group were consistently honest throughout the workshop | 0 | 0 | 0 | 2 | 5 |
| I felt able to talk honestly throughout the session | 0 | 0 | 0 | 3 | 4 |
| I felt comfortable to talk about my preferences | 0 | 0 | 0 | 4 | 3 |
| I felt comfortable to talk about my reasoning for preferences | 0 | 0 | 0 | 6 | 1 |
| The workshop facilitation was appropriate for the content and group | 0 | 0 | 0 | 7 | 0 |

| Workshop | | | | | | |
|---|---|---|---|---|---|---|
| | Strongly disagree | Disagree | Neither agree nor disagree | Agree | Strongly agree | |
| I would recommend this workshop to others | 0 | 0 | 3 | 4 | 0 | |
| If I talk about the workshop to other people it will mostly be positive | 0 | 0 | 1 | 5 | 0 | |
| The outputs of this workshop should be interesting to other audiences | 0 | 0 | 1 | 4 | 0 | |
| | Variable | Very slow | A bit slow | About right | A bit rushed | Too fast |
| The pace of the workshop was | 0 | 0 | 1 | 4 | 0 | 0 |

| Crossover points (Willingness to pay) | | | | | |
|---|---|---|---|---|---|
| | I hadn't really ever thought about it | About the same as I expected | Slightly different from what I expected | Very different from what I expected | |
| On average, other people in the group had preferences that were: | 1 | 5 | 1 | 0 | |
| | Strongly disagree | Disagree | Neither agree nor disagree | Agree | Strongly agree |
| The crossover approach has added something to the way I will think about water investment decisions | 0 | 1 | 1 | 3 | 0 |
| The crossover process helped to inform my thinking about water investment decisions | 1 | 1 | 4 | 1 | 0 |
| The focus on crossover points is a valuable way to guide group discussion | 0 | 2 | 2 | 2 | 0 |

## 4. Discussion

*4.1. Water-Related Insights, Relevant to Other Farmers, Future Irrigation Scheme Design, and Policy Makers*

As expected, the discussion indicated that a range of factors influenced the conditions in which farmers considered an investment in water "worth it". The elicited minimum value generation to make SE3 water worthwhile deviated from the notion that irrigation of traditional annual crops (e.g., cereals) or livestock is not "worth it". This challenges the current focus on marginal gains in informing investment decisions of potential irrigators (as in [57]). What "worth it" means is personal. For some, it involves mainly monetary factors, while for others it is based more on intrinsic motivations, as also acknowledged by, for example, [10,66]. Participants discussed the factors they considered in their decision-making and the meaning and value of these factors as reasons for differences between participants in their WTP between participants. WTP for water may start with dollars, but it does not end there, as the price of water does not reflect all the various intangible aspects within a business structure, for example, in drought-proofing the core of a business. Especially when the decision to invest in water is seen as intergenerational—to allow the next generation to keep on farming—it seems to be less influenced by short-term profits.

The current scheme design strategy in Tasmania has very limited flexibility. Once the irrigation pipes are in place, with the pipe diameters determined by the current demand for summer water, a once in a lifetime opportunity will have passed. The scheme provides no mechanism for further increases in regional water availability during the irrigation season. During the interviews, participants indicated that they hardly ever spoke with other farmers about their reasons for investing (or not). Even during the pre-feasibility stage of the irrigation scheme, when they had to commit to buying water rights, most did not discuss the pros and cons with peers. The insights from this participatory water valuation process provide reasons for pursuing learning by doing and challenge the current approach of building merely for current "demand". A learning-by-doing approach is also recommended by, for example, [41,67]. Based on the participants' experiences and predictions, water demand and WTP are likely to change, as beginner irrigators indicated they had to learn how to use their water, and properties will change ownership over time. The notion that human and water systems evolve in a dynamic learning process is explored in the literature on coupled human-water interactions [68–70], in which a debate is under way about whether and how to deal with unknowns and ambiguity [71,72]. Acknowledgement by policymakers and scheme designers that knowledge and understanding will change in the future raises a water governance challenge to make no-regret design decisions in irrigation infrastructure (see, e.g., [73]). How to co-finance joint irrigation schemes, while acknowledging these changes, provides an interesting avenue for further research on water valuation scenarios.

*4.2. Process Evaluation*

4.2.1. Context Conditions

Our preliminary work flagged the conditions for a dialogue as potentially problematic: the valley had a polarized context in which conditions for a constructive dialogue did not seem to exist beforehand. However, at the workshop, the group dynamics quickly became productive and non-judgmental. Participants felt at ease sharing and admitting not knowing. In the evaluation, participants indicated they had been comfortable talking about their preferences and even about the reasons underlying their preferences. They felt able to talk honestly and believed others were honest as well. These indications and the change in group dynamics suggest that the conditions for a constructive dialogue were created during the workshop. What caused this change is difficult to pinpoint. We provide here a list of possible factors:

1.  The productive environment might have been primed during the introduction to the workshop, (e.g., by showing Table 1) or encouraged by the facilitator [74]. Literature on facilitation observes that a facilitator intentionally and unintentionally influences the process [75–78]. In-depth

knowledge of the area (partly garnered through the interviews) and knowing the participants by name helped the facilitator to provide a safe environment for the participants and to "deepen" the discussion by asking relevant follow-up questions.

2. The relationship initiated during the interviews might have created an incentive to cooperate, in addition to existing power relations between the participants and the chairperson of the SE3 scheme, who made the initial contact.

3. Although the participants were not excessively polite, their behavior might have been influenced by a social code that dictates the need to stay friendly and communicative in a group setting.

4. External factors such as "the right timing" might have influenced the process. The scheme was already in place, which might have made it easier for the irrigators to speak freely rather than try to convince others to buy as well.

### 4.2.2. Effectiveness of Method and Future Work

Given the early, proof-of-concept stage and limited application to one small group, it is impossible to definitively evaluate the utility of WTP scenarios for starting discussion about investment decisions. The evaluation conducted immediately after the workshop and the follow-up phone interviews did, however, provide evidence that this workshop approach is worth investigating further. The participants indicated increased respect for different views, suggesting that judgements and animosities can be reduced by group learning in this format, focused on discussing reasoning and acknowledging subjectivity in the optimum. Trust built through such group learning may provide a foundation for cooperation and collaboration and help create a common understanding of the diverse values among peers. This benefits individuals' capacity to make informed investment decisions on joint infrastructure. There are two paths forward: further applications of the method and deeper investigation of the mechanisms at play.

This study did suggest areas for improvement. Special attention needs to be paid to how individuals in the group prefer to interact, as well as to differences in participants' prior experience. The task of filling in the paper worksheet before and after the discussion was perceived as confusing and distracting. The confidence levels reported either increased or stayed the same; only one participant became less confident about a crossover point on winter water. Explanations of "why" either confidence levels or crossover points changed were very limited as well. As the flow of the discussion is important, we recommend not asking for confidence levels or attempting before-after tests in future applications, unless the workshop is otherwise specifically designed to accommodate them.

The contingent valuation literature acknowledges that survey design, including the questions asked, influences outcomes [28,65,79,80]. In a previous workshop facilitated by the authors, participants compared alternative water sources [39]. In the workshop described here, the questions related to a binary choice of whether or not to invest in water for irrigation. The first three questions related to the cost side, asking participants to indicate their WTP; the last question elicited the value that had to be generated to make the investment worthwhile. An interesting avenue for future research would be to explore whether and how the angle of the questions influences the quantitative responses and discussion. Based on our experience, we recommend keeping the questions simple. Participants then have the opportunity to explain why the question is too simple.

A further promising avenue of research would be to assess the outcomes of such participatory workshops and how the method applied contributed to those outcomes.

## 5. Conclusions

This study presented a method for exploring and sharing in a group setting the reasoning behind decisions to invest. Though the context at first did not seem to be optimal, participants engaged in a productive dialogue, focused first on eliciting individual WTPs and then on exploring the origins of differences between group members. The process encouraged participants to think about the conditions

in which they would (or would not) invest in water. Asking "why?" facilitated a broader discussion of personal reasoning.

Participants observed that valuation of water is a personal matter, as illustrated by the differences in cost prices per ML under which participants invest in water for irrigation. Farmers' personal reasoning went beyond short-term (economic) gains. Indeed, their reasoning did not seem to align with the idea of maximizing short-term profits; it was much more diverse, spanning lifestyle choices, long-term intergenerational planning, perceived risks, and intrinsic motivations. We therefore recommend revising the information provided to potential irrigators, as it currently focuses too strongly on marginal benefits. We also suggest considering peer-to-peer workshops, such as the one described here, to enrich the knowledge of potential water buyers. A better understanding of the personal reasoning of others could provide new insights into investment decision-making processes, improve understanding of differences, and lead to more respect for others and their decisions.

The case presented here illustrates the merit of a participatory valuation method to guide a group discussion about the personal reasoning underlying a WTP for water. In this example, participants had the opportunity to listen, to share their reasoning, to discuss, and to learn from and with others. We demonstrated that discussing reasons underlying an individual WTP is a promising way of helping participants better understand how water is valued, as called for by the UN High Panel on Water [35]. Hence, it can help inform decision-making about joint irrigation infrastructure. Further testing in additional case studies appears to be worthwhile.

**Author Contributions:** Conceptualization, M.J.N. and J.H.A.G.; methodology, M.J.N., J.H.A.G. and P.L.; investigation, M.J.N. and P.L.; writing, original draft preparation, M.J.N.; writing, review, and editing, M.J.N., J.H.A.G. and P.J.G.J.H.; visualization, M.J.N. and J.H.A.G.; supervision, P.L. and P.J.G.J.H.

**Funding:** Joseph Guillaume received financial support from the Academy of Finland funded WASCO project (grant no. 305471) and the Emil Aaltonen Foundation funded project "Eat-Less-Water".

**Acknowledgments:** We thank the participating farmers, who were extremely generous with their time and knowledge. It has been a great pleasure to work with such experts. We also thank Peter Rand, the TI irrigation scheme manager, for providing in-depth scheme information; Neville Mendham for contacting the irrigation committee; and Prof. Art Dewulf for commenting on an early draft. We are grateful for the comments of two anonymous reviewers and the expert copy-editing of Michelle Luijben.

**Conflicts of Interest:** The authors declare no conflict of interest.

## Appendix A. Interview Format

Name                    Date
The interview is set up to explore diversity and to get:
-Better understanding of farming context in the district
-Initial values for irrigation costs
-Initial values for value generated with irrigation water
-Crossover analysis introduced to the participants of the session
**Part 1: Accompanied survey**
**Introduction**
These are questions on your property and water entitlement. With your answers, we are not looking for significance but they will shape the discussion in December.
**General on property**
-How many hectares/acres do you have in total? (1 ac = 0.4 ha)
-How many hectares/acres do you irrigate?
-What crops and pastures do you grow under irrigation?
-On how many hectares or on what area?
**Irrigation water**
-How many years of irrigation experience do you have?
-What is your current water allocation?
-How much water do you use for irrigation per year?

-How does that vary over the years (min/max)?

-How much value do you currently generate per ML?

-How much value do you want to create in the future?

-What is the zone that you are in?

-How tradable is your entitlement? (Location dependent) Have you ever traded water?

-What are your yearly costs for water rights?

-What investments did you have to do to start irrigating (capital cost farm dam, infrastructure)?

-How much does it cost you to put a ML on the ground (ongoing cost)?

**Part 2: Semi-structured interviews to understand context**

**Introduction**

After gathering information about your farm, I would now like to discuss irrigation water in a broader context.

-What was your reasoning for investing (or not) in SE3 water (return on investment)?

-Where did you get information that helped you with this investment decision? What information did you use?

-Do you wish you had bought more/less? Why?

-Do you often discuss water with other farmers? If yes, what aspects?

-What do you think is the long-term reliability of the scheme?

-Where do you see your farm in 20 years' time?

-How do you think the district will develop?

**Introduce crossover and discuss what I want to do in the sessions.**

*At the session I organized in the Coal River Valley, it turned out that participants had very different perspectives on irrigation water and in what conditions it is worth it to invest in water rights. In this district you just had the situation where you actually had to make this investment decision. In the discussion session in December, we will talk about personal reasoning why you did or did not invest in water. Maybe opinions have changed now? We will discuss in what conditions you believe investing in SE3 water is worth it, both based on the costs to buy and operate it and the value you have to generate to make the investment worthwhile.*

**Concluding questions**

Are you willing to participate in the discussion session? Y/N

We are considering the following dates: 4, 5, or 12 December. Do you have a preference?

**Appendix B. Workshop Planning**

18:45 for 19:00 start

Facilitator welcomes everyone when entering by shaking hands (acknowledge). Tea, coffee, and Dutch cookies are provided near the entrance so participants can get comfortable and have something to do. Next to the coffee and tea, there will be stickers to make name tags.

19:05

Facilitator starts by thanking everybody for coming and explains the rules/conditions without being directive or demanding: *The interesting part of tonight is talking about reasons behind outcomes and personal preferences. My findings so far suggest that everyone here has personal reasons for what they do. These reasons define personal preferences and investment decisions. Tonight, there is no "best". There is no stupid, or smart. There is no better or worse. Diversity in this group is its biggest asset. Diversity is the key to new insights. I hope that you are curious and open to others. I hope that you are willing to listen and learn so that you leave with new insights, and I hope you are willing to explain and share so that others can learn from you. I will first display some ranges of the answers that I got in the interviews. These interview findings show differences between you, but do not explain where these differences come from. That is what you will discuss in the rest of the session. Is that clear? Are there any questions?*

Clicker question (CQ)

CQ How many of the other participants have you discussed water with?

Would you please introduce yourself: Name, type of enterprise, location of farm?

CQ How confident do you feel discussing costs and benefits of irrigation water in this group?

Is there anything we can do here and now to improve the situation before we start discussing?

Display:

Range of: value generated/ML, years of experience, setup to start with, ML/ha, crops grown, cost to put water on crops (pumping)

Explain the crossover concept and introduce the steps:

First guess, confidence levels, second guess, confidence levels -> discuss changes.

Water-related crossover questions:

*From the interviews, I learned that farm location is important. The proximity to residential areas made some of you prefer keeping the option of subdivision open. That is a very important insight and I will report on that, but we will have to take that out of the consideration today. Therefore, today we explore the costs and benefits of irrigation water for a farming future. We first go back in time a bit. Based on what you know now, if the scheme was going to be rebuilt and there was not the option of subdividing your land in the future, what would you have done?*

CQ 1) Wat is the maximum price for a water right that you would be willing to pay per ML?

Follow up      - Is that different now than two years ago?

         - Is there anyone here who wished they had bought more, or less? Why? Who bought extra since? Why?

Reliability came up in the interview as a very important characteristic. Let's explore that, and again, I am interested in the reasons why:

CQ 2) How much would you have been willing to pay if reliability would have been 80% instead of 95%? Why?

Follow up      - Would you have bought more? Why?

         - How important is reliability? Why?

         - How can management affect reliability?

         - What if you could order in bulk? Why?

20:00–20:15 Break: Coffee, tea and Grolsch

CQ 3) How much are you willing to pay for winter water rights? Why?

Follow up:      - Manageability

         - What are the most important differences between winter water and summer water?

         - Will non-irrigators buy winter water?

         - Is there enough water to facilitate the valley's long-term potential?

CQ 4) How much value/ML do you have to generate to make SE3 water worthwhile? Why?

Follow up:      - Is that possible with livestock? Annual crops? Why? How?

         - What are other factors that should be considered? Why?

            (Succession)

            (Long-term investment)

            (Value of water security, risk, uncertainty)

            (More value generation but not more profit?)

            (Change in life: Working at night and in the weekends)

20:45 Evaluation

21:15–21:30 Wrap up

## Appendix C. Workshop Evaluation: Follow-Up Evaluation Questions

1. Do you remember the purpose of the workshop? Were there any parts of the discussion that stood out or that you remember as particularly useful or interesting?

2. Are there any ways that you think of that the crossover process could be adapted or improved to make it more useful or to achieve its full potential?

3. Would you recommend the workshop to others? If so, why?

4. Did the discussion give you a better understanding of, or confidence in, your preferences? If so, can you say what this influence is?
5. Any other things you would like to add?

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
