# Peer review of "Sharing Reasoning Behind Individual Decisions to Invest in Joint Infrastructure"

_water, doi:10.3390/w11040798_

Round 1

Reviewer 1 Report

Overall paper can be improved. Please, find my comments with attached file.

Author Response

Please find our responses in the attached word file

Reviewer 2 Report

The manuscript is very well written. Points are sound and clear. I have some minor questions to discuss.

I suggest, in the introduction and discussion section, adding some background information on the historical hydrology events that caused losses to farmers, which

can be possibly relaxed by having new infrastructures. The possible future climate change is another factor that needs to be considered when building new infrastructures.

These information can be communicated with farmers to have more interactive discussions such that their WTP may be altered.  Comparing to government or other large organizations input on building new infrastructures, the possible percentage contributed by farmers needs to be evaluated.  Will their input play a significant role or not? Some quantifications may be needed to further justify the value of the work. Of course, these numbers would be different for different study sites.

Author Response

please find our responses in the attached word file
